# CAN LARGE LANGUAGE MODELS INFER CAUSATION FROM CORRELATION?

**Zhijing Jin**[1,2,*,‡]   **Jiarui Liu**[3,*]   **Zhiheng Lyu**[4]   **Spencer Poff**[5]
**Mrinmaya Sachan**[2]   **Rada Mihalcea**[6]   **Mona Diab**[3,‡,†]   **Bernhard Schölkopf**[1,†]
[1]Max Planck Institute for Intelligent Systems, Tübingen, Germany   [2]ETH Zürich
[3]LTI, CMU   [4]University of Hong Kong   [5]Meta AI   [6]University of Michigan
jinzhi@ethz.ch   jiarui@cmu.edu   zhihenglyu.cs@gmail.com

## ABSTRACT

Causal inference is one of the hallmarks of human intelligence. While the field of Causal NLP has attracted much interest in the recent years, existing causal inference datasets in NLP primarily rely on discovering causality from empirical knowledge (e.g., commonsense knowledge). In this work, we propose the first benchmark dataset to test the pure causal inference skills of large language models (LLMs). Specifically, we formulate a novel task CORR2CAUSE, which takes a set of correlational statements and determines the causal relationship between the variables. We curate a large-scale dataset of more than 200K samples, on which we evaluate seventeen existing LLMs. Through our experiments, we identify a key shortcoming of LLMs in terms of their causal inference skills, and show that these models achieve almost close to random performance on the task. This shortcoming is somewhat mitigated when we try to re-purpose LLMs for this skill via finetuning, but we find that these models still fail to generalize – they can only perform causal inference in in-distribution settings when variable names and textual expressions used in the queries are similar to those in the training set, but fail in out-of-distribution settings generated by perturbing these queries. CORR2CAUSE is a challenging task for LLMs, and can be helpful in guiding future research on improving LLMs' pure reasoning skills and generalizability.[1]

## 1 INTRODUCTION

Causal inference, i.e., the ability to establish the correct causal relationships between variables or events, is fundamental to human intelligence. There are two distinct ways this causal inference capability can be acquired: one through empirical knowledge, e.g., we know from common sense that touching a hot stove will get us burned; the other through *pure causal reasoning*, as causality can be formally argued and reasoned about using known procedures and rules from causal inference (Spirtes et al., 2000; Pearl, 2009; Peters et al., 2017). One example is that we have the a priori knowledge that the correlation between A and B does not necessarily imply causality. This is a formal rule that holds true regardless of the realizations of the variables A and B.

With the rise of large language models (LLMs) (Radford et al., 2019; Devlin et al., 2019; Ouyang et al., 2022; Zhang et al., 2022; OpenAI, 2023, *inter alia*), a crucial research question is whether they can do causal reasoning well. Recent studies have pointed out that LLMs are "causal parrots," which recite the causal knowledge in the training data (Zečević et al., 2023). Moreover, the vast majority of studies frame causal reasoning as a skill to navigate around empirical knowledge (Gordon et al., 2012; Sap et al., 2019a;b; Qin et al., 2019; Bhagavatula et al., 2020), and also treat LLMs as a knowledge base when evaluating its causal skills (Kıcıman et al., 2023; Tu et al., 2023; Xie et al., 2023). However, all the above lines of research frame causality as empirical knowledge, thus relying heavily on the quality and the coverage of the training data, overlooking the great potential of the formal causal reasoning skills to process correlational information to causal conclusions.

---

*Equal contribution. †Equal supervision. ‡Work originated as a Meta AI internship project involving Zhijing, Mona, and Spencer.

[1]Our data is at https://huggingface.co/datasets/causalnlp/corr2cause.
Our code is at https://github.com/causalNLP/corr2cause.

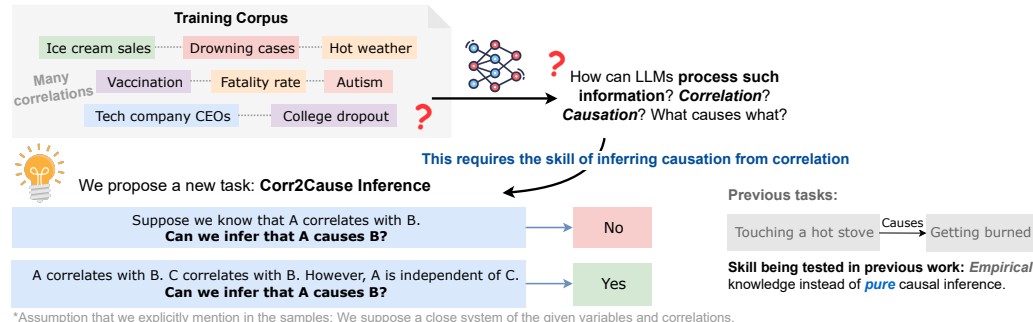

Figure 1: Illustration of the motivation behind our task and dataset.

Drawing inspirations from technical studies on causal discovery (Spirtes et al., 2000; Spirtes & Zhang, 2016; Glymour et al., 2019), we formulate a novel task for NLP, *correlation-to-causation inference* (CORR2CAUSE), which is an important skill for LLMs. Imagine the scenario in Figure 1, where the training corpus does not tediously cover every causal relation, but more pervasively talk about correlations, such as which events tend to co-occur. Learning a good CORR2CAUSE skill can enable LLMs to draw causal relations behind the mere correlational information on the surface. For example, several decades ago, there might be an observation that female university students tend to perform better, but behind the correlational statistics is the causal graph that female students have to achieve extra good performance to get into universities as the first place.

To this end, we collect the CORR2CAUSE dataset, the first dataset to test the pure causal reasoning abilities of LLMs. All the questions in this dataset are centered around testing when it is valid or invalid to infer causation from correlation. To systematically compose this dataset, we ground our generalization process in the formal framework of causal discovery (Spirtes et al., 1993; 2000; Glymour et al., 2016; Spirtes & Zhang, 2016), which provides rules about how to deduce causal relations among variables given their statistical correlation in the observational data. We generate more than 200K data points, and label a correlation-causation statement pair as valid if and only if there is a bijective mapping between the statistical correlation and the underlying causality.

Based on our CORR2CAUSE dataset with 200K samples, we investigate two main research questions: (1) How well do existing LLMs perform on this task? (2) Can existing LLMs be re-trained or re-purposed on this task and obtain robust causal inference skills? Through extensive experiments, we show empirically that none of the 17 existing LLMs we investigate perform well on this pure causal inference task. We also show that although LLMs can demonstrate better performance after being finetuned on the data, the causal inference skills attained by them are not robust. In summary, our contributions are as follows:

1. We propose the novel task of CORR2CAUSE, to probe an aspect of LLM's reasoning ability, *pure causal inference*;
2. We compose a dataset of over 200K samples, using insights from causal discovery;
3. We evaluate the performance of 17 LLMs on our dataset, finding that all of them perform poorly, close to the random baseline;
4. We further explored whether LLMs can learn the skill through finetuning, and find that LLMs fail to robustly acquire this skill in out-of-distribution settings. Finally, we suggest future work to explore more ways to enhance the pure causal inference skill in LLMs.

## 2 PRELIMINARIES: CAUSAL INFERENCE

### 2.1 DIRECTED GRAPHICAL CAUSAL MODELS (DGCMS)

A directed graphical causal model (DGCM) is a commonly used representation to express the causal relations among a set of variables. Given a set of $N$ variables $\boldsymbol{X} = \{X_1, \ldots, X_N\}$, we can encode the causal relations among them using a directed graph $\mathcal{G} := (\boldsymbol{X}, \boldsymbol{E})$, where $\boldsymbol{E}$ is the set of directed edges. Each edge $e_{i,j} \in \boldsymbol{E}$ represents a causal link $X_i \to X_j$, meaning that $X_i$ is a direct cause of $X_j$. In the context of this work, we take the common assumption of directed acyclic graphs (DAGs), which most causal discovery methods use (Glymour et al., 2019), as graphs with cycles can make the causal discovery process arbitrarily hard.

Following the graph-theoretic terminology, we use an analogy of the ancestry tree to denote the relations between two variables. For example, we call $X_i$ as a *parent* of $X_j$ if there is a directed edge $X_i \rightarrow X_j$ in the graph, and, thus, $X_j$ is a *child* of $X_i$. Similarly, we denote $X_i$ as an *ancestor* of $X_j$ if there exists a directed path from $X_i$ to $X_j$, and, thus, $X_j$ is a *descendent* of $X_i$. Note that a parent is a special case of an ancestor where the directed path has a length of 1.

For convenience, we also introduce the notions for some special three-variable relations. Given two variables $X_i$ and $X_j$, we call a third variable $X_k$ a *confounder* (i.e., *common cause*) if $X_k$ is a parent of both $X_i$ and $X_j$; a *collider* (i.e., *common effect*) if $X_k$ is a child of both $X_i$ and $X_j$; and a *mediator* if $X_k$ is both a child of $X_i$, and a parent of $X_j$.

## 2.2 D-Separation and Markov Property

**D-Separation** D-separation (Pearl, 1988) is a fundamental concept in graphical models used to determine whether two sets of nodes $X$ and $Y$ in a DAG $\mathcal{G}$ are conditionally independent given a third set of nodes $Z$, where the three sets are disjoint. We say that $X$ and $Y$ are d-separated by $Z$ if all paths between any node in $X$ and any node in $Y$ are *blocked* by the conditioning set $Z$. A path between $X$ and $Y$ is blocked by $Z$ if there exists a node $A \in Z$ which satisfies one of the following conditions: $A$ is the parent node in a fork structure on the path (i.e., $\cdot \leftarrow A \rightarrow \cdot$); $A$ is the mediator node in a chain structure on the path (i.e., $\cdot \rightarrow A \rightarrow \cdot$); or in any collider structure on the path (i.e., $\cdot \rightarrow A \leftarrow \cdot$), $Z$ does not contain $A$ or its descendants.

**Markov Property** The Markov property in a DAG $\mathcal{G}$ states that each node $X_i$ is conditionally independent of its non-descendants given its parents, namely $X_i \perp\!\!\!\perp \mathbf{NonDe}(X_i) | \mathbf{Pa}(X_i)$, where $\mathbf{NonDe}(X_i)$ denotes the non-descendants of $X_i$ excluding itself, and $\mathbf{Pa}(X_i)$ denotes the parents of $X_i$. Using the Markov property, we can factorize the joint distribution of all the nodes in the graph into $P(X_1, \ldots, X_N) = \prod_{i=1}^{N} P(X_i | \mathbf{PA}(X_i))$. To infer the causal graph from probability distributions, a common assumption is faithfulness, namely the validity to infer all the d-separation sets in the graph from the independence relations in the probability distribution. In our work, we also take this broadly taken assumption which holds for most real-world scenarios.

**Markov Equivalence of Graphs** We denote two DAGs as Markov equivalent if they induce the same joint distribution $P(X)$. The set of DAGs that are Markov equivalent to each other is called a Markov equivalence class (MEC). Causal graphs in the same MEC can be easily identified since they have the same skeleton (i.e., undirected edges) and V-structures (i.e., structures in the form of $A \rightarrow B \leftarrow C$ where $A$ and $C$ are not connected).

Obviously, there is a one-to-many mapping (i.e., surjection) between the causal graph and statistical distribution. Namely, each causal graph sufficiently determines a statistical distribution, but from a statistical distribution, we cannot necessarily induce a unique causal graph. This is why we say "correlation does not necessarily mean causation".

## 2.3 Causal Discovery

Causal discovery aims to learn the causal relations by analyzing statistical properties in the observational data (Spirtes et al., 1993; 2000; Glymour et al., 2016; Spirtes & Zhang, 2016; Glymour et al., 2019). It can be achieved through constraint-based methods (Spirtes et al., 2000), score-based methods (Chickering, 2002), or other methods taking advantage of the functional causal models (Shimizu et al., 2006; Hoyer et al., 2008; Zhang & Hyvärinen, 2009).

To fit for the spirit of this paper to infer from correlation (expressed in natural language) to causation, we base our dataset design on the widely-used Peter-Clark (PC) algorithm (Spirtes et al., 2000). The PC algorithm is based on the principles of conditional independence and the causal Markov assumption, which allows it to efficiently identify causal relationships among variables in a given dataset. The algorithm first starts with a fully connected undirected graph among all the variables. Then it removes the edge between two variables if there is an unconditional or conditional independence relationship between them. Afterwards, it orients the directed edges whenever there is a V-structure. And finally, it iteratively checks the direction of the other edges until the entire causal graph is consistent with all the statistical correlations.

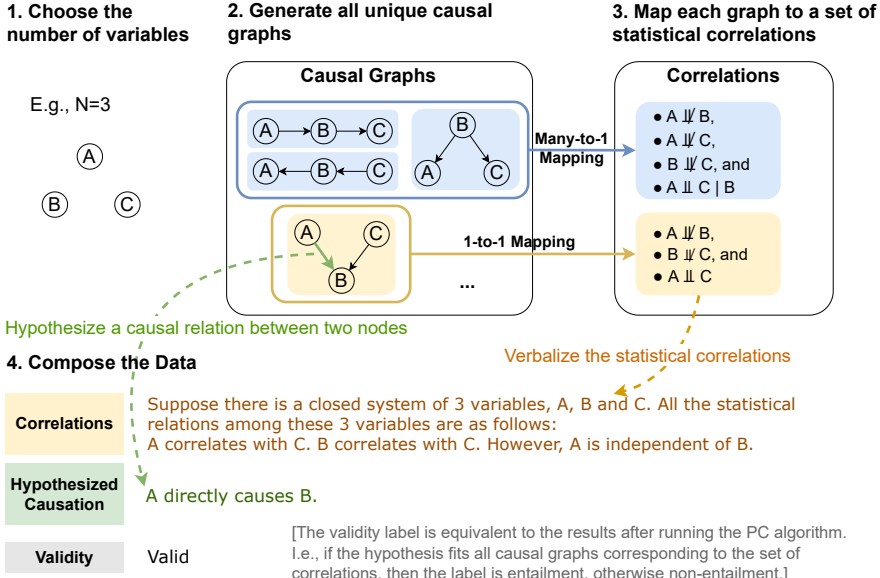

Figure 2: Pipeline of the data construction process.

## 3 DATASET CONSTRUCTION

We introduce the construction of our dataset in this section. We start with our task formulation for CORR2CAUSE, and then briefly give an overview of the data generation process, followed by detailed descriptions of each step. We conclude the section with the overall statistics of the dataset.

### 3.1 TASK FORMULATION

Given a set of $N$ variables $\boldsymbol{X} = \{X_1, \ldots, X_N\}$, we have a statement $\boldsymbol{s}$ about all the correlations among the variables, and a hypothesis $\boldsymbol{h}$ describing the causal relation $r$ between the pair of variables $X_i$ and $X_j$. The task is to learn a function $f : (\boldsymbol{s}, \boldsymbol{h}) \mapsto v$ which maps the correlation statement $\boldsymbol{s}$ and the causal relation hypothesis $\boldsymbol{h}$ to their validity $v \in \{0, 1\}$, which takes the value 0 if this inference is invalid, and the value 1 if this inference is valid.

### 3.2 OVERVIEW OF THE DATA GENERATION PROCESS

We base the construction our dataset on several concepts of causal inference, including the DGCM, d-separation, and MECs, as introduced in Section 2.

As in the overview of our data generation process in Figure 2, we first choose the number $N$ of variables (Step 1) and generate all the unique DGCMs with $N$ nodes (Step 2), which we will introduce in the Section 3.3. Then we collect all the d-separation sets from these graphs to identify MECs (Step 3) in Section 3.4. Then, in Step 4, we create the formal form of data in Section 3.5. For each correspondence of the MEC to causal graphs, we compose the correlation statement based on the statistical relations in the MEC, and hypothesize a causal relation between two variables, and produce the validity $v = 1$ if the hypothesis is a shared property of all causal graphs in the MEC, and $v = 0$ if the hypothesis is not necessarily true for all the MEC graphs. Finally, we introduce the verbalization process in Section 3.6.

### 3.3 CONSTRUCTING THE GRAPHS WITH ISOMORPHISM CHECKS

The first step of the data generation is to compose the causal graphs, as in Step 1 and 2 of Figure 2. For a set of $N$ variables $\boldsymbol{X} = \{X_1, \ldots, X_N\}$, there are $N(N-1)$ possible directed edges, since each node can link to any node other than itself. To remove cycles in the graph, we make the nodes in topological order, which only allows edges $X_i \to X_j$, where $i < j$. We achieve this by limiting the adjacency matrix of the graph to only having non-zero values above the diagonal, resulting in $N(N-1)/2$ possible directed edges for the DAGs.

At the first glance, for $N$ nodes, there should be $2^{N(N-1)/2}$ possible DAGs (i.e., the power set of all edges). However, there could be isomorphic graphs in this set. To avoid this, we perform a graph

| # Nodes | # Unique DAGs | # Edges/DAG | # MECs | # DAGs/MEC |
|---------|---------------|-------------|--------|------------|
| 2 | 2 out of 2 | 0.50 | 2 | 1.0 |
| 3 | 6 out of $2^3$ | 1.67 | 5 | 1.2 |
| 4 | 31 out of $2^6$ | 3.48 | 20 | 1.55 |
| 5 | 302 out of $2^{10}$ | 5.89 | 142 | 2.13 |
| 6 | 5,984 out of $2^{15}$ | 8.77 | 2,207 | 2.71 |
| Total | 6,325 | 8.60 | 2,376 | 2.66 |

Table 1: Statistics about the source causal graphs in our dataset. Given the number of nodes, we report the number of unique DAGs, average number of edges per DAG, number of MECs, and average number of DAGs per MEC.

isomorphism check (McKay & Piperno, 2014), and reduce the set so that only unique DAGs are retained, and we show their statistics in Table 1. Although we can handle large graphs, we mostly focus on smaller graphs that can still lead to a reasonably sized dataset, so we empirically set $N = 6$, but future work can use our open-sourced codes to extend to more nodes.

### 3.4 PROGRAMMATICALLY GENERATING THE D-SEPARATION SETS

Based on the set of unique DAGs, we then programmatically generate the d-separation sets by graph theoretical conditions, as in Step 3 of Figure 2. To realize this step, we code an efficient graph-theoretic algorithm to check for all the chain, fork, and collider structures to automatically identify the set of nodes that d-separate each pair of nodes. Using the d-separation sets and the faithfulness assumption, we form the statistical correlations as follows. For each pair of nodes, they are conditionally independent given the variables in the d-separation set. If the d-separation set is empty, then the two nodes are unconditionally independent. If no d-separation set can be found for the two nodes, then they are directly correlated.

Moreover, using the d-separation sets, we are able to cluster causal graphs to MECs. We achieve it by tracing the mapping between the causal graphs and the set of statistical correlations, and backtracking the graphs with the same d-separation sets to group them in the same MEC. We show in Table 1 that each MEC contains on average 2.66 DAGs.

### 3.5 COMPOSING THE HYPOTHESES AND LABEL

After generating the set of correlations based on the d-separation sets, we now generate the causal hypotheses. For the causal relation $r$, we focus on six common causal relations between two nodes introduced in Section 2.1: Is-Parent, Is-Child, Is-Ancestor (excluding the parents), Is-Descendant (excluding the children), Has-Confounder (i.e., there exists a confounder, or common cause, of the two nodes), and Has-Collider (i.e., there exists a collider, or common effect, of the two nodes). In this way, the set of hypotheses contains all six meaningful causal relations between every pair of variables, resulting in a total size of $6 \cdot N(N-1)/2 = 3N(N-1)$ hypotheses for a graph with $N$ variables.

To generate the ground-truth validity label, we start from the correlation sets in Step 3, then look up all the causal graphs in the same MEC corresponding to the given set of correlations, and check the necessity of the hypothesized causal relation. If the causal relationship proposed in the hypothesis is valid for all causal graphs within the MEC, then we generate the validity $v = 1$; otherwise, we generate $v = 0$. A special case of valid samples is that when the size of the MEC is 1, then there is a bijective mapping between the causal graph and the d-separation sets, so any hypothesis stating the causal properties of that unique causal graph is valid.

### 3.6 VERBALIZING INTO LANGUAGE

Finally, as in the last step of Figure 2, we convert all the information above to text data for our CORR2CAUSE task. For the correlation statement, we verbalize the set of correlations in Step 3 into a natural language statement $s$. When two variables cannot be d-separated, i.e., $A \not\perp\!\!\!\perp B$, then we describe them as "$A$ correlates with $B$" since they are directly correlated and cannot be independent by any condition. And if two variables have a valid d-separation set $C$, then we describe them as "$A$ is independent of $B$ given $C$." In the special case when the d-separation set is empty, we directly say "$A$ is independent of $B$." In addition, we disambiguate the setting by starting the correlation statement with the setup of a closed system of the given variables, and no hidden variables: "Suppose there is a closed system of $N$ variables, A, B, ... All the statistical relations among these $N$ variables are as follows:". Finally, to verbalize the hypothesis, we feed the causal relation triplet $(X_i, r, X_j)$

| Causal Relation | Hypothesis Template |
|---|---|
| Is-Parent | `{Var i}` directly causes `{Var j}`. |
| Is-Ancestor | `{Var i}` causes something else which causes `{Var j}`. |
| Is-Child | `{Var j}` directly causes `{Var i}`. |
| Is-Descendant | `{Var j}` is a cause for `{Var i}`, but not a direct one. |
| Has-Collider | There exists at least one collider (i.e., common effect) of `{Var i}` and `{Var j}`. |
| Has-Confounder | There exists at least one confounder (i.e., common cause) of `{Var i}` and `{Var j}`. |

Table 2: Templates for each causal relation in the hypothesis. We use `{Var i}` and `{Var j}` as placeholders for the two variables.

into their hypothesis templates in Table 2. For example, we turn the triplet $(A, \text{Is-Parent}, B)$ into "$A$ directly causes $B$", as in the example of Figure 2.

## 3.7 STATISTICS OF THE RESULTING DATA

We show the statistics of our CORR2CAUSE dataset in Table 3. Overall, our dataset contains 207,972 samples, where 18.57% of the samples have the positive label (i.e., with validity=1). The average length of the premise is 424.11 tokens, and hypothesis 10.83 tokens. We split the data into 205,734 training samples, 1,076 development and 1,162 test samples.[2] Since the main purpose of the dataset is to benchmark the performance of LLMs, we prioritize the test and development sets to have a comprehensive coverage over all sizes of graphs. Specifically, we iterate through the subset of our data for each $N$, and split it entirely for only the test and development sets if the data is less than 1K, which is the case for $N = 2$ and 3. For the other subsets that are larger, we randomly sample up to 1K or 10% of the data, whichever is smaller, to the test and development sets. We set the cap to be 1K in order to form a reasonable computation budget, since many LLMs are expensive to query in the inference mode. Aside from the test and valid sets, all the rest of the data goes into the training set.

| | Overall | Statistics by the Number of Nodes $N$ | | | | |
|---|---|---|---|---|---|---|
| | | $N = 2$ | $N = 3$ | $N = 4$ | $N = 5$ | $N = 6$ |
| # Samples | 207,972 | 12 | 90 | 720 | 8,520 | 198,630 |
| # Test | 1,162 | 6 | 48 | 72 | 514 | 522 |
| # Dev | 1,076 | 6 | 42 | 72 | 482 | 474 |
| # Train | 205,734 | 0 | 0 | 576 | 7,524 | 197,634 |
| # Tokens/Premise | 424.11 | 31.5 | 52.0 | 104.0 | 212.61 | 434.54 |
| # Tokens/Hypothesis | 10.83 | 10.83 | 10.83 | 10.83 | 10.83 | 10.83 |
| % Positive Labels | 18.57 | 0.00 | 3.33 | 7.50 | 13.01 | 18.85 |
| Vocab Size | 65 | 49 | 53 | 55 | 57 | 61 |

Table 3: Statistics of our CORR2CAUSE dataset, and by subsets. We report the total number of samples (# Samples); splits of the test (# Test), developement (# Dev) and training sets (# Train); number of tokens per premise (# Tokens/Premise) and hypothesis (# Tokens/Hypothesis); percentage of the positive labels (% Positive Labels), and vocabulary size by the number of unique tokens (Vocab Size). Note that the number of unique graphs and MECs are in Table 1.

## 4 EXPERIMENTS

### 4.1 EXPERIMENTAL SETUP

We set up a diverse list of LLMs for the experiments on our CORR2CAUSE dataset. To *test existing LLMs*, we first include six commonly used BERT-based NLI models in the transformers library (Wolf et al., 2020): BERT (Devlin et al., 2019), RoBERTa (Liu et al., 2019), BART (Lewis et al., 2020), DeBERTa (He et al., 2021), DistilBERT (Sanh et al., 2019), and DistilBART (Shleifer & Rush, 2020). Apart from these BERT-based NLI models, we also evaluate the general-purpose autoregressive LLMs based on GPT (Radford et al., 2019): GPT-3 Ada, Babbage, Curie, Davinci (Brown et al., 2020); its instruction-tuned versions (Ouyang et al., 2022), text-davinci-001, text-davinci-002, and text-davinci-003; and GPT-3.5 (i.e., ChatGPT), and the latest GPT-4 (OpenAI, 2023) by April 2023,

---

[2]*Note for our dataset v2.0:* We notice that our original data (v1.0) has duplication due to symmetric relations and verbalizations of the hypothesis. E.g., Is-Parent(A, B) has the exact hypothesis verbalization as Is-Child(B, A). Hence, for our data v2.0, we perform a careful de-duplication, and update the data statistics in Table 3. See more version comparison details in Appendix D. Note that, due to the symmetry, the current version is a random sample half of the size of the original version, so the modeling results in the experiment section roughly hold.

|  | F1 | Precision | Recall | Accuracy |
|---|---|---|---|---|
| ***Random Baselines*** | | | | |
| Always Majority | 0.0 | 0.0 | 0.0 | 84.77 |
| Random (Proportional) | 13.5 | 12.53 | 14.62 | 71.46 |
| Random (Uniform) | 20.38 | 15.11 | 31.29 | 62.78 |
| ***BERT-Based Models*** | | | | |
| BERT MNLI | 2.82 | 7.23 | 1.75 | 81.61 |
| RoBERTa MNLI | 22.79 | 34.73 | 16.96 | 82.50 |
| DeBERTa MNLI | 14.52 | 14.71 | 14.33 | 74.31 |
| DistilBERT MNLI | 20.70 | 24.12 | 18.13 | 78.85 |
| DistilBART MNLI | 26.74 | 15.92 | 83.63 | 30.23 |
| BART MNLI | **33.38** | 31.59 | 35.38 | 78.50 |
| ***LLaMa-Based Models*** | | | | |
| LLaMa-7B | 26.81 | 15.50 | 99.42 | 17.36 |
| Alpaca-7B | 27.37 | 15.93 | 97.37 | 21.33 |
| ***GPT-Based Models*** | | | | |
| GPT-3 Ada | 0.00 | 0.00 | 0.00 | 84.77 |
| GPT-3 Babbage | 27.45 | 15.96 | 97.95 | 21.15 |
| GPT-3 Curie | 26.43 | 15.23 | 100.00 | 15.23 |
| GPT-3 Davinci | 27.82 | 16.57 | 86.55 | 31.61 |
| GPT-3 Instruct (text-davinci-001) | 17.99 | 11.84 | 37.43 | 48.04 |
| GPT-3 Instruct (text-davinci-002) | 21.87 | 13.46 | 58.19 | 36.69 |
| GPT-3 Instruct (text-davinci-003) | 15.72 | 13.4 | 19.01 | 68.97 |
| GPT-3.5 | 21.69 | 17.79 | 27.78 | 69.46 |
| GPT-4 | 29.08 | 20.92 | 47.66 | 64.60 |

Table 4: Overall performance. We report F1 (main metric), precision, recall and accuracy. For the main metric, F1 score, we use the **bold** font to highlight the overall best performance, and underline to highlight the best performance within each category of models.

using the OpenAI API (`https://openai.com/api/`) with temperature 0. We also evaluate the recent, more efficient models, LLaMa (Touvron et al., 2023) and Alpaca (Taori et al., 2023).

When inspecting the behavior of *finetuned models*, we adopt a large set of models, including GPT-based models (GPT-3 Ada, Babbage, Curie, and Davinci) using the OpenAI finetuning API for classification at `https://platform.openai.com/docs/guides/fine-tuning`, open-sourced decoder-only models (GPT2, GPT2-Large, GPT2-XL, LLaMA-7B, and LLaMA2-7B), BERT-based models from scratch (BERT-Base, BERT-Large, RoBERTa-Base, and RoBERTa-Large), and BERT-Based NLI models (BERT-Base MNLI, BERT-Large MNLI, RoBERTa-Base MNLI, and RoBERTa-Large MNLI) using the transformers library (Wolf et al., 2020). See training details in Appendix A.

For the *random baselines*, we provide "always majority" to predict the majority class 100% of the time, "random (uniform)" to uniformly sample a label (i.e., 50% for each), and "random (proportional)" to sample a label from a Bernouli distribution proportional to the development set label distribution.

## 4.2 THE CORR2CAUSE SKILL IN EXISTING LLMS

We show the performance of seventeen LLMs in Table 4. We can see that pure causal inference is a very challenging task across all existing LLMs. Among all the LLMs, the best performance is 33.38% F1 by BART MNLI, which is even higher than the latest GPT-based model, GPT-4. Notably, many models are worse than random guess, which means that they totally fail at this pure causal inference task. The observation still holds for few-shot chain-of-thought prompts tested in Appendix G.

## 4.3 FINETUNED PERFORMANCE

Next, we address the question: *Can we re-purpose LLMs to learn this task?* The experimental results in Table 5a of 17 models finetuned on our CORR2CAUSE seem very strong at first sight. Most models see a substantial increase, among which the finetuned BERT-based NLI models demonstrate the strongest performance. The best-performing one, RoBERTa-Large MNLI, achieves 94.74% F1 score on this task, as well as very high precision, recall and accuracy scores.

| | F1 | Precison | Recall | Accuracy | | F1 (Paraph.) | F1 (Var. Ref.) |
|---|---|---|---|---|---|---|---|
| ***Finetuned GPT-Based Models Using OpenAI API*** | | | | | | | |
| GPT-3 Ada | 79.85 | 70.47 | 92.11 | 92.92 | | 61.73 | 41.57 |
| GPT-3 Babbage | 78.19 | 69.98 | 88.60 | 92.48 | | 62.34 | 43.28 |
| GPT-3 Curie | 81.23 | 75.00 | 88.60 | 93.77 | | 64.93 | 45.32 |
| GPT-3 Davinci | 85.52 | 80.26 | 91.52 | 95.28 | | 65.01 | 46.96 |
| ***Finetuned Open-Sourced Decoder-Only Models*** | | | | | | | |
| GPT2 | 89.18 | 88.03 | 90.35 | 96.66 | | 56.76 | 31.70 |
| GPT2-Large | 94.29 | 92.18 | 96.49 | 98.22 | | 55.95 | 31.99 |
| GPT2-XL | 94.30 | 91.94 | 96.78 | 98.22 | | 60.32 | 43.95 |
| LLaMA-7B | 91.98 | 88.62 | 95.61 | 97.46 | | 56.41 | 53.92 |
| LLaMA2-7B | 92.92 | 90.11 | 95.91 | 97.77 | | 52.24 | 49.47 |
| ***Finetuned BERT-Based Models*** | | | | | | | |
| BERT-Base | 69.29 | 54.42 | 95.32 | 87.13 | | 61.13 | 35.20 |
| BERT-Large | 85.26 | 77.51 | 94.74 | 95.01 | | 63.64 | 38.54 |
| RoBERTa-Base | 87.60 | 78.47 | 99.12 | 95.73 | | 65.58 | 53.12 |
| RoBERTa-Large | 89.10 | 82.54 | 96.78 | 96.39 | | 65.05 | 60.20 |
| ***Finetuned BERT-Based NLI Models*** | | | | | | | |
| BERT-Base MNLI | 89.88 | 85.49 | 94.74 | 86.51 | | 65.56 | 31.50 |
| BERT-Large MNLI | 90.19 | 84.44 | 96.78 | 96.79 | | 67.24 | 52.04 |
| RoBERTa-Base MNLI | 94.27 | 90.35 | 98.54 | 98.17 | | 57.42 | 62.83 |
| RoBERTa-Large MNLI | **94.74** | 92.24 | 97.37 | 98.35 | | 55.45 | 67.87 |

(a) Performance of finetuned models on the original test set.

(b) F1 scores of finetuned models on the perturbed test sets by paraphrasing (Paraph.) and variable refactorization (Var. Ref.).

Table 5: Performance of finetuned models on the original test set and perturbed test sets.

| Relation Type | F1 | Precision | Recall | Accuracy | | F1 | Precision | Recall | Accuracy |
|---|---|---|---|---|---|---|---|---|---|
| Is-Parent | 96.18 | 95.45 | 96.92 | 98.67 | | 74.80 | 79.31 | 70.77 | 91.73 |
| Is-Ancestor | 93.94 | 93.94 | 93.94 | 98.93 | | 45.45 | 90.91 | 30.30 | 93.60 |
| Is-Child | 95.73 | 94.92 | 96.56 | 98.67 | | 73.39 | 78.43 | 68.97 | 92.27 |
| Is-Descendant | 96.55 | 93.33 | 100 | 99.47 | | 29.41 | 83.33 | 17.86 | 93.60 |
| Has-Collider | 92.19 | 87.41 | 97.52 | 94.64 | | 70.70 | 75.00 | 66.90 | 82.04 |
| Has-Confounder | 98.67 | 97.37 | 100 | 99.73 | | 70.42 | 73.53 | 67.57 | 94.37 |

(a) Fine-grained performance of RoBERTa-Large by causal relation type on the original test set.

(b) Its fine-grained performance by relation type after variable refactorization.

Table 6: Fine-grained analysis of the best-performing model, RoBERTa-Large MNLI.

## 4.4 FINE-GRAINED PERFORMANCE BY CAUSAL RELATION

In addition to the overall results mentioned above, we conduct a fine-grained analysis to check the performance of the strongest finetuned model, RoBERTa-Large MNLI, by our six causal relation types. As in Table 6a, the model is very good at judging relations such as Is-Parent, Is-Descendant and Has-Confounder, all with more than 96% F1 scores, whereas it is several points weaker on the Has-Collider relations. This could be due to that the collider relation is the most special type, requiring identification of the V-structure based on both the unconditional independence based on the two variables only and correlations whenever conditioned on a common descendant. We also conduct error analysis for non-finetuned models in Appendix F.

## 4.5 ROBUSTNESS ANALYSIS

Looking at the very high performance of the finetuned models, we raise the next question: *Did the models really robustly learn the causal inference skills?*

**Two Robustness Tests** We design two simple robustness tests: (1) paraphrasing, and (2) variable refactorization. For (1) paraphrasing, we simply paraphrase the hypothesis by changing the text template for each causal relation to some semantically-equivalent alternatives in Appendix C. For (2) variable refactorization, we reverse the alphabet of the variable names, namely flipping A, B, C, to Z, Y, X and so on. The inspiration behind the two robustness tests comes from the spurious correlation analysis described in Appendix E.

Specifically, we adopt the common setup of text adversarial attack (Morris et al., 2020; Jin et al., 2020) to preserve the training set and keep the same saved models, but run the inference on the perturbed test set. In this way, we separate the possibilities of the models only overfitting on the training data vs. mastering the reasoning skills.

**Results after Perturbation** We can see from Table 5b that all the models drop drastically, by up to 39.29 on the paraphrased test set, and up to 62.30 after variable refactorization. The best-performing model, RoBERTa-Large MNLI, is especially sensitive towards paraphrasing, demonstrating the most drop among all models; however, it is the most robust against the variable refactorization, maintaining a high F1 score of 67.87. We conduct fine-grained analysis for RoBERTa-Large MNLI under perturbation in Table 6b. We can see the the main source of the performance drop of the model comes from the two classes, Is-Ancestor (decreasing to 45.45%) and Is-Descendant (decreasing to 29.41%), while the other classes stay relatively robust, keeping their F1 scores over 70%.

From this analysis, we make the following suggestions to future studies testing this CORR2CAUSE skill of LLMs. First, it is safe to use it as a test set to benchmark existing LLMs' performance, since the data we generate is out-of-distribution from the training data of the current LLMs. Then, when testing finetuned models, it is very important to accompany adversarial attack together with the i.i.d. test set. We open-source our perturbed test sets for future work to test the generalizability skill.

## 4.6 EXTENSION TO NATURAL STORIES

We envision our CORR2CAUSE dataset to be a foundation for future extensions to various settings, such as instantiating the variables with actual phenomena and situating the story in a more natural setting. For example, the *correlation does not imply causation* rule can be instantiated with the ice cream sales and swimming pool attendance as the two variables, and argue that ice cream sales does not necessarily affect swimming pool attendance, because their correlation could be due to a third variable, such as hot weather. We provide a case study for how to instantiate the symbolic expressions in our dataset to more natural stories, and find that LLMs such as GPT-4 can generate realistic, daily life stories that has foreseeably broad applications. See more details in Appendix B.

## 5 RELATED WORK

**Existing Causal Reasoning Tasks** A large body of existing research of causal reasoning in NLP focuses on leveraging empirical knowledge to do tasks such as inferring the cause and effect of why an agent perform certain tasks (Sap et al., 2019a), the motivation and emotional reaction in a social context (Sap et al., 2019b), how people achieve a given goal with a set of concrete steps (Zhang et al., 2020), the development of a story given a different beginning (Qin et al., 2019), and how in general LLMs serve as a knowledge base of cause and effect (Willig et al., 2023; Kıcıman et al., 2023). In contrast, our CORR2CAUSE task focuses on the pure causal inference skill of models, which is a knowledge-dependent reasoning skill based on formally correct rules from causal inference.

**Existing Logical and Inference Tasks** Another related area of literature is logical and inference tasks, of which a well-established one is natural language inference (NLI), to identify the semantic relationship between a pair of sentences (MacCartney & Manning, 2008; Bowman et al., 2015). NLI datasets mainly focus on the set and paraphrase relations. For example, "a group of boys are playing football" can entail "some guys are playing football," where "boys" are a sub-concept of "guys," and "a group of" and "some" are paraphrases. Recently, there have been increasing efforts to extend the inference task to various logical inference skills such as deductive logic and propaganda techniques (Jin et al., 2022; Alhindi et al., 2022). Our CORR2CAUSE dataset is the first dataset testing the correlation-to-causation inference skill, which is unique of its type.

## 6 CONCLUSION

In this work, we introduced a novel task, CORR2CAUSE, to infer causation from correlation, and collected a large-scale dataset of over 200K samples. We evaluated an extensive list of LLMs on this new task, and showed that off-the-shelf LLMs perform poorly on this task. We also show that it is possible to re-purpose LLMs on this task by finetuning, but future work needs to be aware of the out-of-distribution generalization problem. To avoid the Goodhart's law, we recommend using this dataset to benchmark the pure causal inference skills for LLMs that have not seen this dataset. Given the limited reasoning abilities of current LLMs, and the difficulty of separating actual reasoning from training-corpus-derived knowledge, it is imperative that our community focus on work aiming to accurately disentangle and measure both abilities. We believe the present work is a first such step.

## LIMITATIONS AND FUTURE WORK

We identify several limitations of this work and open future directions: First, in the context of this work, we limit the causal graphs to two to six nodes, but future work can feel free to explore larger graphs. Another aspect is that we do not assume hidden confounders in this inference problem, so we welcome future work to generate an even more challenging dataset to infer the existence of hidden confounders, analogous to the causal discovery algorithm of fast causal inference (FCI) (Spirtes et al., 2000). And also in general, explorations of other causal discovery algorithms are welcomed too. Finally, a lot of motivation behind proposing this task is inspired by the problem of invalid reasoning patterns in our daily reasoning (Jin et al., 2022), which could fertilize the ground for more pervasive spread of fake news. We believe false causal inference is a prevalent type of fallacious beliefs, and welcome future work to connect the idea of this benchmark to more real-world false beliefs based on confusing correlation with causation.

## ACKNOWLEDGMENT

We thank Riley Goodside for valuable suggestions to improve our prompts to LLMs. We thank Luigi Gresele and Amir Hossein Karimi for their suggestions to help us improve the formulation of our causal discovery questions.

This material is based in part upon work supported by the German Federal Ministry of Education and Research (BMBF): Tübingen AI Center, FKZ: 01IS18039B; by the Machine Learning Cluster of Excellence, EXC number 2064/1 – Project number 390727645; by a National Science Foundation award (#2306372); by a Swiss National Science Foundation award (#201009) and a Responsible AI grant by the Haslerstiftung. Zhijing Jin is supported by PhD fellowships from the Future of Life Institute and Open Philanthropy. We also thank OpenAI for granting Zhijing quota to their API of GPT series through the Researcher Access Program.

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

## A    IMPLEMENTATION DETAILS

When finetuning on our data, for GPT-based models, we use the default settings of the OpenAI finetuning API; and for BERT-based models, we use the `transformers` library (Wolf et al., 2020) and train the models on a server with an NVIDIA Tesla A100 GPU with 40G of memory. To fit for the GPU memory, we set the batch size to be 8. We use the validation set to tune the learning rate, which takes value in {2e-6, 5e-6, 1e-5, 2e-5, 5e-5}; dropout rate, which takes value in {0, 0.1, 0.2, 0.3}; and weight decay, which takes value in {1e-4, 1e-5}. We train the models until convergence, which is usually around ten epochs.

**Prompts** When querying the autoregressive LLMs, we formulate the prompt as follows:

*Question:* `[premise]`

*Can we deduct the following:* `[hypothesis]`*? Just answer "Yes" or "No."*

*Answer:*

## B    GENERATING NATURAL STORIES

To generate the natural stories based on our symbolic expressions, we utilize the state-of-the-art LLM, GPT-4, which is very good at story generation. We design detailed instructions in the prompt, and generate around 200 stories in our case study. We show two examples stories in Table 7, and the report the overall statistics in Table 8.

|  | Example 1 (Label=Negative) | Example 2 (Label=Positive) |
|---|---|---|
| Symbolic Form | *Premise:* Suppose there is a closed system of 2 variables, A and B. All the statistical relations among these 2 variables are as follows: A correlates with B.
*Hypothesis:* A directly affects B.
*Relation between the promise and hypothesis:* The premise does not a necessary condition for the hypothesis. | *Premise:* Suppose there is a closed system of 3 variables, A, B and C. All the statistical relations among these 3 variables are as follows: A correlates with C. B correlates with C. However, A is independent of B.
*Hypothesis:* A directly affects C.
*Relation between the promise and hypothesis:* The premise is a necessary condition for the hypothesis. So if the premise is true, the hypothesis must be true. |
| Natural Story | *Premise:* Suppose there is a closed system of 2 variables, ice cream sales and swimming pool attendance. All the statistical relations among these 2 variables are as follows: ice cream sales correlate with swimming pool attendance.
*Hypothesis:* Ice cream sales directly affect swimming pool attendance.
*Relation between the premise and hypothesis:* The premise does not provide a necessary condition for the hypothesis. The correlation between ice cream sales and swimming pool attendance could be due to a third variable, such as hot weather, which increases both ice cream sales and swimming pool attendance. Therefore, it is not necessarily true that ice cream sales directly affect swimming pool attendance. | *Premise:* Let's consider three factors: eating junk food (A), obesity (C), and watching television (B). There is a correlation between eating junk food and obesity, and between watching television and obesity. However, eating junk food and watching television are independent from each other.
*Hypothesis:* Eating junk food directly affects obesity.
*Relation between the premise and hypothesis:* The premise provides the necessary conditions for the hypothesis. It establishes the independent variables A (eating junk food) and B (watching television) and their correlations with obesity. Given that these are true, it supports the hypothesis that eating junk food directly affects obesity. |

Table 7: Examples of natural stories generated based on the symbolic form in our CORR2CAUSE dataset, showing the broad application value of our dataset as the starting point for various verbalizations of the correlation-to-causation inference task.

| | |
|---|---|
| Test Set Size | 102 |
| Dev Set Size | 102 |
| # Tokens/Premise | 64.88 |
| # Tokens/Hypothesis | 13.54 |
| # Tokens/Explanation | 64.66 |
| % Positive Labels | 1.67 |

Table 8: Statistics of our generated natural stories. We report the number of samples in the test and development sets; number of tokens per premise (# Tokens/Premise), hypothesis (# Tokens/Hypothesis), and explanation (# Tokens/Explanation); and percentage of the positive labels (% Positive Labels).

For more information, the exact prompt we use is "*Here is a causal inference rule: `[symbolic form]` Please provide a real-world example instantiating this phenomenon. Format it also as "Premise:", "Hypothesis:", and "Relation between the promise and hypothesis:".*"

## C  TEMPLATES AND PARAPHRASES

We use the verbalization templates in Table 9 to compose the hypotheses for all six causal relations.

| Causal Relation | Hypothesis Template |
|---|---|
| Is-Parent | {Var i} directly causes {Var j}. |
| Is-Ancestor | {Var i} causes something else which causes {Var j}. |
| Is-Child | {Var j} directly causes {Var i}. |
| Is-Descendant | {Var j} is a cause for {Var i}, but not a direct one. |
| Has-Collider | There exists at least one collider (i.e., common effect) of {Var i} and {Var j}. |
| Has-Confounder | There exists at least one confounder (i.e., common cause) of {Var i} and {Var j}. |
| *Paraphrases* | |
| Is-Parent | {Var i} directly affects {Var j}. |
| Is-Ancestor | {Var i} influences {Var j} through some mediator(s). |
| Is-Child | {Var j} directly affects {Var i}. |
| Is-Descendant | {Var j} influences {Var i} through some mediator(s). |
| Has-Collider | {Var i} and {Var j} together cause some other variable(s). |
| Has-Confounder | Some variable(s) cause(s) both {Var i} and {Var j}. |

Table 9: Templates and their paraphrases for each causal relation in the hypothesis. We use {Var i} and {Var j} as placeholders for the two variables.

## D  CHANGE LOG FOR THE DATASET VERSION UPDATE

| Two Equivalent Forms | Duplication Property | De-Duplication Method |
|---|---|---|
| Is-Parent(i, j)
Is-Child(j, i) | Two exact same strings | Keep only one, by forcing $i < j$ |
| Is-Ancestor(i, j) (Original)
Is-Descendent(j, i) (Original) | Two different strings, but semantically equivalent | Randomly sample one out of the two |
| Is-Ancestor(i, j) (Paraphrased)
Is-Descendent(j, i) (Paraphrased) | Two exact same strings | Keep only one, by forcing $i < j$ |
| Has-Collider(i, j)
Has-Collider(j, i) | Two different strings, but semantically equivalent | Randomly sample one out of the two |
| Has-Confounder(i, j)
Has-Confounder(j, i) | Two different strings, but semantically equivalent | Randomly sample one out of the two |

Table 10: De-duplication methods for the six causal relation types and their verbalizations.

**De-Duplication Strategy** As mentioned in Section 3.7 in the main paper, our original dataset (v1.0) has duplication due to symmetric relations and verbalizations. We introduce in Table 10 several reasons for why duplicated hypotheses exist in our original data. One typical reason is symmetric relations such as Is-Parent(A, B) and Is-Child(B, A), and, similarly, the paraphrased version of

Is-Ancestor(A, B) and Is-Descendent(B, A). Another typical reason is the semantic equivalence in the verbalization templates, which applies to the Has-Collider and Has-Confounder relations. For example, the verbalized texts of Has-Collider(A, B) and Collider(B, A) are "There exists at least one collider (i.e., common effect) of {A and B, B and A}," respectively, which are semantically-equivalent paraphrases of each other, so we randomly keep one out of the two.

**Resulting Dataset Statistics after De-Duplication**

Since the reason for duplication in the first place is due to symmetry in the causal relation, or verbalization, the resulting new data, CORR2CAUSE v2.0, is exactly a half of the original data. As we reported previously in Table 3 of Section 3.7, the total number of samples cuts down to half, while the label distribution and all other properties are the same. To compose each split, we apply the same de-duplication method for the test, train, and development sets. We notice that some duplicates are across the splits, so we prioritize keeping the test and training sets untouched (to minimally affect the experimental results), and then reduce the development set by removing the cross-split duplicates, namely:

- test_2.0 = deduplicate(test_1.0)
- train_2.0 = deduplicate(train_1.0)
- dev_2.0 = deduplicate(dev_1.0) \ {test_2.0, train_2.0}

We expect minimal or almost no change to the experimental results. In case of the slight possibility that this change in the development set might affect the model selection in the training process, future work can feel free to re-train the models and update the exact performance number.

# E  SPURIOUS CORRELATION ANALYSIS

The inspirations of our two robustness tests (paraphrasing and variable refactorization) come from our data analysis. We check for spurious correlations in the data by reporting in Table 11 the point-wise mutual information (PMI) between the label and any n-gram with no more than four tokens. In addition, we also report the difference of the PMI with the two labels in the |Diff| column of Table 11, and report the top 10 n-grams.

The design spirit for our robustness test is that if the models' correct judgment relies on exploiting these spurious correlations, then such reliance will be broken in our perturbations.

| N-Gram | PMI w/ Non-Ent. Label | PMI w/ Ent. Label | \|Diff\| |
|---|---|---|---|
| a cause | 1.692209 | -1.025611 | 2.717820 |
| a cause for | 1.663640 | -0.983790 | 2.647430 |
| A causes | 1.640679 | -0.951610 | 2.592289 |
| A causes something | 1.621820 | -0.926075 | 2.547895 |
| a direct | 1.606052 | -0.905316 | 2.511369 |
| a direct one | 1.592673 | -0.888107 | 2.480781 |
| for D | 1.584826 | -0.878180 | 2.463006 |
| for D but | 1.583897 | -0.877014 | 2.460911 |
| for E | 1.582980 | -0.875864 | 2.458844 |
| for E but | 1.582074 | -0.874728 | 2.456802 |

Table 11: PMI between the labels and n-grams. The labels include non-entailment (Non-Ent.) and entailment (Ent.). And the n-grams include all with no more than four words. The |Diff| column shows the absolute value of the difference between the PMIs with two labels. We show the top 10 n-grams with the largest differences of their PMIs with the two classes in the |Diff| column.

We can see that some spurious correlations are rooted in the framing of the hypothesis, such as "a cause (for)", and "a direct (one)" (which we use the paraphrasing task to break), and others are connected to the variable names, such as "for D (but)" and "for E (but)" (which we use the variable refactorization to break).

# F  FINE-GRAINED ERROR ANALYSIS

In addition to the fine-grained analysis by causal relation type in Table 6a for fine-tuned models, we also report such error analysis for non-finetuned models in Table 12.

| Selected Models | Relation Type | F1 | Precision | Recall | Accuracy |
|---|---|---|---|---|---|
| GPT-3.5 | All | 21.69 | 17.79 | 27.78 | 69.46 |
| GPT-3.5 | Is-Parent | 8.82 | 100 | 4.62 | 83.47 |
| GPT-3.5 | Is-Ancestor | 0 | 0 | 0 | 90.67 |
| GPT-3.5 | Is-Child | 9.84 | 100 | 5.17 | 85.33 |
| GPT-3.5 | Is-Descendant | 14.29 | 11.9 | 17.86 | 84 |
| GPT-3.5 | Has-Collider | 34.24 | 25.51 | 52.07 | 35.12 |
| GPT-3.5 | Has-Confounder | 15.33 | 8.86 | 56.76 | 37.8 |
| GPT-4 | All | 29.08 | 20.92 | 47.66 | 64.6 |
| GPT-4 | Is-Parent | 0 | 0 | 0 | 82.67 |
| GPT-4 | Is-Ancestor | 30.77 | 31.25 | 30.3 | 88 |
| GPT-4 | Is-Child | 0 | 0 | 0 | 84.53 |
| GPT-4 | Is-Descendant | 26.98 | 17.35 | 60.71 | 75.47 |
| GPT-4 | Has-Collider | 44.1 | 30.18 | 81.82 | 32.71 |
| GPT-4 | Has-Confounder | 20.67 | 11.53 | 100 | 23.86 |
| RoBERTa MNLI | All | 22.79 | 34.73 | 16.96 | 82.5 |
| RoBERTa MNLI | Is-Parent | 0 | 0 | 0 | 82.67 |
| RoBERTa MNLI | Is-Ancestor | 0 | 0 | 0 | 91.2 |
| RoBERTa MNLI | Is-Child | 0 | 0 | 0 | 84.53 |
| RoBERTa MNLI | Is-Descendant | 0 | 0 | 0 | 92.53 |
| RoBERTa MNLI | Has-Collider | 43.45 | 39.73 | 47.93 | 59.52 |
| RoBERTa MNLI | Has-Confounder | 0 | 0 | 0 | 84.45 |

Table 12: Fine-grained evaluation results for some selected non-fine-tuned models.

These results are particularly revealing, showing how off-the-shelf models perform in recognizing specific relations. Specifically, GPT-3.5 cannot recognize ancestor relations, whereas GPT-4 fails at all direct causation recognition with parents and children. And RoBERTa MNLI only did collider relation relatively correctly. Note that, when the F1 score is zero, the accuracy number is a result of always predicting the negative class of that relation.

## G  LLM PERFORMANCE OPTIMIZATION

Since our experiments in Section 4.2 are based on plain, zero-shot prompts, we explore whether better prompting strategies could improve the performance. We enhance the query prompt by incorporating several strategies: (1) Utilizing a system prompt that specifies the model's expertise ("You are a highly intelligent question-answering bot with profound knowledge of causal inference."); (2) Including a pair of few-shot examples, one positive and one negative; (3) Implementing chain-of-thought prompting with "Let's think step by step." to encourage the language model to generate step-by-step reasoning. In Table 13, we present the evaluation results on the relatively affordable model, GPT-3.5, where the optimized prompt leads to a 4-point improvement in F1 over the original performance. However, we can see that despite the deployment of all three strategies, the model continues to struggle with this challenging task.

| | F1 | Precision | Recall | Accuracy |
|---|---|---|---|---|
| GPT-3.5 (plain query; original) | 21.69 | 17.79 | 27.78 | 69.46 |
| GPT-3.5 (enhanced query) | 25.44 | 17.29 | 48.11 | 52.01 |

Table 13: Performance of GPT-3.5 with different queries. We quote the original performance from Table 4.

