# OpenReview forum: "Can Large Language Models Infer Causation from Correlation?"
_ICLR.cc/2024/Conference — ICLR 2024 poster_

### Official Review · Reviewer_fGug · 2023-10-31

**Soundness:** 3 good
**Presentation:** 3 good
**Contribution:** 3 good
**Rating:** 6
**Confidence:** 4

**Summary:**

To investigate the capability of llm on inferring causality from correlations in text data, this paper proposed a novel corr2cause benchmark dataset. Apart from introducing the detailed dataset construction process, this paper also conducted extensive experiments over 17 llms to verify the effectiveness of the dataset. Meanwhile, this paper also verified whether llm can learn the causality skill after finetuning. In conclusion, this paper made an interesting attempt over casual inference ability of llms.

**Strengths:**

1.	This paper design a novel corr2cause dataset to verify the causal inference ability of LLMs.
2.	This paper provided a detailed dataset construction introduce, which makes the dataset very reasonable.
3.	This paper conducted extensive experiments over 17 LLMs. Moreover, further experiments were also conducted to verify whether LLMs can learn causal inference ability by finetuning. The experiments are very convincing.
4.	This paper made an early and interesting attempt over the causal inference ability of LLMs, which can inspire plenty of related work.

**Weaknesses:**

1.	For the dataset construction process, the authors leverage variables to represent the input data. However, what do variables represent? Entity in sentence? sentences or what? The authors should provide more details information.
2.	For the causality discovery, the authors only provided the results of PC algorithm. However, the precision of PC algorithm is unclear. Whether the results are reliable is still unclear. The author should provide more details about how to ensure the confidence of the results generated by PC algorithm.

**Questions:**

N/A

---

> ### Author Response · Authors · 2023-11-23
>
> Thank you for your positive feedback on our paper, particularly for recognizing the novelty of our dataset, the detailed technical process of data construction, the extensive experimentation with 17 language models, and our in-depth further experiments and analysis.
>
> **Reviewer's Query on Dataset Construction Process:**
>
> > The dataset construction process utilizes variables to represent the input data. Could you clarify what these variables signify? Do they represent entities within sentences, the sentences themselves, or something else? More detailed information would be beneficial.
>
> **Response:**
> Thank you for raising this important question. We provided some examples in our “Appendix B: Generating Natural Stories.” In our data, variables are symbolically named and can be verbalized to represent entities, such as “eating junk food (A), obesity (C), and watching television (B),” and the inference problem can be
>
> *Premise: Let’s consider three factors: eating junk food (A), obesity (C), and watching television (B). There is a correlation between eating junk food and obesity, and between watching television and obesity. However, eating junk food and watching television are independent from each other. Hypothesis: Eating junk food directly affects obesity.*
>
> Note that our current Corr2Cause dataset primarily employs symbolic forms to distill reasoning patterns (which adheres to our motivation to improve pure causal reasoning). With this piece of solid technical work, we provide good potential for future work to instantiate the variables using various real-world entities, facilitating the generation of natural stories that align with practical applications, such as *combating misinformation*.
>
>
>
> **Reviewer's Query on Causality Discovery:**
>
> > The paper only discusses results obtained using the PC algorithm for causality discovery. The precision of this algorithm is not clearly stated, raising questions about the reliability of the results. Could you provide more details on how you ensure the confidence in the results generated by the PC algorithm?
>
> **Response:**
> Thank you for the question. We selected the PC algorithm for its suitability in processing inputs formatted as correlations. Traditionally there are two main types of methods for causal discovery: constraint-based methods (based on the dependence and independence relations among variables), and score-based methods (which takes in the data distribution, and rank the possible causal graphs by their scores given the data, e.g., Bayesian Information Criterion (BIC) score). In our dataset, we made a design choice that natural language tests usually do not explicitly contain data samples to run statistical tests on (which is where the confidence scores come from), but instead they typically contain *observations of which variables are related or independent*. Therefore, we design our task as the Corr2Cause inference task, which looks at the core task of inferring causal conclusions based on the presented correlations, and bypasses the step of statistical testing from raw data samples. For this task with correlations as input, constraint-based methods, specifically the most classic PC algorithm, is a great fit. As the dependent and independences are provided, there will not be statistical tests to produce confidence scores, but only algorithmic reasoning to identify the causal graphs deterministically.
>
> The remaining major requirement of the PC algorithm is the faithfulness assumption, which we introduce in the preliminaries section. We adhere to this faithfulness assumption, which is generally met in real-world scenarios, although exceptions can occur if the causal mechanism is intricately designed to obscure statistical dependencies. We appreciate your attention to this matter. We will make sure to emphasize this point again in the limitations section, and suggest future work exploring different causal discovery algorithms.
>
>
>
> We appreciate your time and feedback on our paper. Feel free to let us know anytime if you have further questions.

---

### Official Review · Reviewer_WuGy · 2023-11-01

**Soundness:** 3 good
**Presentation:** 4 excellent
**Contribution:** 3 good
**Rating:** 6
**Confidence:** 4

**Summary:**

This work proposes a new task called Corr2Cause. Specifically, the input example specifies the correlation and independence of a multi-variable closed system, and the model is asked to judge the causal relationships inside this system. For this task, this work constructs a dataset containing more than 400k examples for 2-6 variable systems. 17 models are evaluated on this task, including BERT-style NLI models, and GPT-style auto-regressive LLMs. None of the models perform well out of the box. While fine-tuning BERT-style NLI models can bring significant improvements, these improvements cannot be generalized to paraphrased examples or examples with different variable names. Finally, this work also presents a case study generating another version of the dataset with natural stories using GPT4.

**Strengths:**

1. Differentiating causality from correlations is an important task and it is one of the clear limitations of LLMs.
2. The description of the construction process of the dataset is detailed, well-written, and sound. The background on basic causal inference can be useful for general readers.
3. The evaluation covers a wide range of models. Both traditional BERT-style NLI models and GPT-style LLMs are evaluated for this task.

**Weaknesses:**

1. I'm concerned about the difficulty of this task. This task can be very difficult for the models because it also involves multi-step reasoning, understanding symbolic systems, knowing the definition of all the terminologies, etc. Despite the clear motivation of evaluating the model's ability to do causal discovery, for me, it is unclear which part is actually the bottleneck of this task. From the current evaluation results, I am convinced that the models cannot do multi-step reasoning to figure out causality from a symbolic system, but I cannot make more fine-grained claims.
2. It's not a major point and I don't know if the authors have enough computation resources. But while the generalization experiment design is neat, it is not surprising that BERT-style models do not generalize well to paraphrases and new variables. The evaluation would be more interesting if the authors could fine-tune some relatively small-scale GPT-style models for this experiment.
3. In order to have good coverage for the dev and test set, this work removes all the 2-3 variable examples from the training set. Removing simple examples can make compositional tasks like the task for this work very hard. I wonder if this design makes this dataset unnecessarily difficult, and whether such a dataset split is the best configuration for this task.

**Questions:**

1. Have you done any sub-sampling for the dataset construction? Or have you put all the possible examples and hypotheses in the dataset? This part is not clear to me.
2. Do you have fine-grained evaluation results (like Table 6) for non-fine-tuned models? I find Table 6 to be very informative.

---

> ### Author Response · Authors · 2023-11-23
>
> Thank you for your positive comments on our paper, particularly for recognizing the value of the task in distinguishing causality from correlations, detailed technical processes we used to construct the dataset, and the comprehensive evaluation in the experiments.
>
> > Have you done any sub-sampling for the dataset construction? Or have you included all possible examples and hypotheses in the dataset? This aspect was not entirely clear to me.
>
> In constructing the Corr2Cause dataset, we included all possible examples and hypotheses. This comprehensive approach ensures that our dataset captures the full spectrum of inference tasks of all combinations of correlations among the given set of variables.
>
> > Do you have fine-grained evaluation results (like Table 6) for non-fine-tuned models? I found Table 6 particularly informative.
>
> Thank you for this constructive suggestion. In response to your request, we have obtained fine-grained evaluation results for non-fine-tuned models in the following table:
>
> | Model        | Relation Type  | F1    | P     | R     | Acc   |
> | ------------ | -------------- | ----- | ----- | ----- | ----- |
> | GPT-3.5      | All            | 21.69 | 17.79 | 27.78 | 69.46 |
> | GPT-3.5      | Is-Parent      | 8.82  | 100   | 4.62  | 83.47 |
> | GPT-3.5      | Is-Ancestor    | 0     | 0     | 0     | 90.67 |
> | GPT-3.5      | Is-Child       | 9.84  | 100   | 5.17  | 85.33 |
> | GPT-3.5      | Is-Descendant  | 14.29 | 11.9  | 17.86 | 84    |
> | GPT-3.5      | Has-Collider   | 34.24 | 25.51 | 52.07 | 35.12 |
> | GPT-3.5      | Has-Confounder | 15.33 | 8.86  | 56.76 | 37.8  |
> |              |                |       |       |       |       |
> | GPT-4        | All            | 29.08 | 20.92 | 47.66 | 64.6  |
> | GPT-4        | Is-Parent      | 0     | 0     | 0     | 82.67 |
> | GPT-4        | Is-Ancestor    | 30.77 | 31.25 | 30.3  | 88    |
> | GPT-4        | Is-Child       | 0     | 0     | 0     | 84.53 |
> | GPT-4        | Is-Descendant  | 26.98 | 17.35 | 60.71 | 75.47 |
> | GPT-4        | Has-Collider   | 44.1  | 30.18 | 81.82 | 32.71 |
> | GPT-4        | Has-Confounder | 20.67 | 11.53 | 100   | 23.86 |
> |              |                |       |       |       |       |
> | RoBERTa MNLI | All            | 22.79 | 34.73 | 16.96 | 82.5  |
> | RoBERTa MNLI | Is-Parent      | 0     | 0     | 0     | 82.67 |
> | RoBERTa MNLI | Is-Ancestor    | 0     | 0     | 0     | 91.2  |
> | RoBERTa MNLI | Is-Child       | 0     | 0     | 0     | 84.53 |
> | RoBERTa MNLI | Is-Descendant  | 0     | 0     | 0     | 92.53 |
> | RoBERTa MNLI | Has-Collider   | 43.45 | 39.73 | 47.93 | 59.52 |
> | RoBERTa MNLI | Has-Confounder | 0     | 0     | 0     | 84.45 |
>
>
>
>
>
> These results are particularly revealing, showing how off-the-shelf models perform in recognizing specific relations. Specifically, GPT-3.5 cannot recognize ancestor relations, whereas GPT-4 fails at all direct causation recognition with parents and children. And RoBERTa MNLI only did collider relation relatively correctly. Note that when the F1 score is zero, the positive accuracy results from predicting the negative class of that relation.
>
> > In order to ensure comprehensive coverage for the development and test sets, this work excludes all 2-3 variable examples from the training set. I'm concerned that this decision may render the dataset overly challenging and question whether such a split is optimal for this task.
>
> Thank you for highlighting this concern. Our objective was to balance the coverage of the training set with the comprehensiveness of the test and development sets. When faced with trade-offs due to the limited number of 2-3 variable examples, we prioritized the quality of the test and development sets. As we discussed in the paper, even fine-tuned models exhibit limitations in robust causal reasoning, as it might fall into spurious correlations. Therefore, we believe that the test set serving as unseen data for LLMs well have more value than putting things into the training data and making models memorize them.
>
> On a separate note, we understand that this is a challenging task, and perhaps to mitigate this issue, future research may opt to use the 2-3 variable cases in the development set for few-shot prompting, or reallocate the 2-3 variable data samples between the training and development sets to optimize the performance.
>
>
> > Despite the clear motivation behind evaluating the model's capability in causal discovery, I find it challenging to pinpoint the exact bottleneck of this task.
>
> Thank you for your insightful comment. Following your suggestions, we are conducting a more fine-grained error analysis by utilizing chain-of-thought prompting to explicitly generate and examine the step-by-step reasoning of LLMs, and we're manually annotating different types of reasoning errors in a detailed manner. This analysis is currently in progress, and we plan to share the results either in this discussion thread, or directly later in the camera-ready version.

---

> ### Author Response · Authors · 2023-11-23
>
> > It's not a critical issue, but I'm curious if the authors had sufficient computational resources for this. While the generalization experiment design is commendable, it's unsurprising that BERT-style models do not generalize well to paraphrases and new variables. The evaluation could be more compelling if you could fine-tune some smaller-scale GPT-style models for this experiment.
>
> We appreciate your insightful suggestion and your consideration of our computational resources. During the rebuttal period, we managed to conduct experiments with small-scale GPT-style models, including gpt2 (124M parameters), gpt2-large (774M), gpt2-xl (1.5B). We further enriched the results by including the latest LLaMa series of models, LLaMa-7B and LLaMa2-7B, for which we run the LoRA versions which are more lightweight.
>
> First, we show the overall performance results below. We find that for each model series, the larger or later ones achieve better results, and overall GPT2-XL after fine-tuning (gpt2-xl-ft) achieves the best performance, on par with the fine-tuned Roberta model's 94.74% F1.
>
> | Model         | F1        | Precision | Recall | Accuracy |
> | ------------- | --------- | --------- | ------ | -------- |
> | gpt2-ft       | 89.18     | 88.03     | 90.35  | 96.66    |
> | gpt2-large-ft | 94.29     | 92.18     | 96.49  | 98.22    |
> | gpt2-xl-ft    | **94.30** | 91.94     | 96.78  | 98.22    |
> | llama-ft      | 91.98     | 88.62     | 95.61  | 97.46    |
> | llama2-ft     | 92.92     | 90.11     | 95.91  | 97.77    |
>
>
> Additionally, inspired by your previous suggestion, we obtained fine-grained performance data for these models. Due to space constraints, we are presenting a selection from the series of GPT-2 models:
>
> | Model         | Relation Type  | F1    | P     | R     | Acc   |
> | ------------- | -------------- | ----- | ----- | ----- | ----- |
> | gpt2-ft       | All            | 89.18 | 88.03 | 90.35 | 96.66 |
> | gpt2-ft       | Is-Parent      | 91.04 | 88.41 | 93.85 | 96.8  |
> | gpt2-ft       | Is-Ancestor    | 90.91 | 90.91 | 90.91 | 98.4  |
> | gpt2-ft       | Is-Child       | 95    | 91.94 | 98.28 | 98.4  |
> | gpt2-ft       | Is-Descendant  | 87.72 | 86.21 | 89.29 | 98.13 |
> | gpt2-ft       | Has-Collider   | 82.64 | 82.64 | 82.64 | 88.74 |
> | gpt2-ft       | Has-Confounder | 97.3  | 97.3  | 97.3  | 99.46 |
> |               |                |       |       |       |       |
> | gpt2-large-ft | All            | 94.29 | 92.18 | 96.49 | 98.22 |
> | gpt2-large-ft | Is-Parent      | 95.52 | 92.75 | 98.46 | 98.4  |
> | gpt2-large-ft | Is-Ancestor    | 95.65 | 91.67 | 100   | 99.2  |
> | gpt2-large-ft | Is-Child       | 94.12 | 91.8  | 96.55 | 98.13 |
> | gpt2-large-ft | Is-Descendant  | 98.25 | 96.55 | 100   | 99.73 |
> | gpt2-large-ft | Has-Collider   | 93.88 | 92.74 | 95.04 | 95.98 |
> | gpt2-large-ft | Has-Confounder | 89.47 | 87.18 | 91.89 | 97.86 |
> |               |                |       |       |       |       |
> | gpt2-xl-ft    | All            | 94.3  | 91.94 | 96.78 | 98.22 |
> | gpt2-xl-ft    | Is-Parent      | 95.52 | 92.75 | 98.46 | 98.4  |
> | gpt2-xl-ft    | Is-Ancestor    | 92.75 | 88.89 | 96.97 | 98.67 |
> | gpt2-xl-ft    | Is-Child       | 94.92 | 93.33 | 96.55 | 98.4  |
> | gpt2-xl-ft    | Is-Descendant  | 94.92 | 90.32 | 100   | 99.2  |
> | gpt2-xl-ft    | Has-Collider   | 93.98 | 91.41 | 96.69 | 95.98 |
> | gpt2-xl-ft    | Has-Confounder | 93.15 | 94.44 | 91.89 | 98.66 |
>
> We can see from above that the collider and descendent relations are relatively challenging for smaller GPT-2 models, and for the largest GPT-2 model, gpt2-xl-ft, it achieves good performance at all causal relations.
>
>
>
> Finally, we conducted two robustness tests on these models, including Variable Refactoring (VR) and Paraphrasing (P). The results are as follows:
>
> | Model            | F1    | Precision | Recall | Accuracy |
> | ---------------- | ----- | --------- | ------ | -------- |
> | gpt2-ft-P        | 56.76 | 57.10     | 56.43  | 86.91    |
> | gpt2-large-ft-P  | 55.95 | 56.97     | 54.97  | 86.82    |
> | gpt2-xl-ft-P     | 60.32 | 52.25     | 71.35  | 85.71    |
> | llama-ft-P       | 56.41 | 55.00     | 57.89  | 86.38    |
> | llama2-ft-P      | 52.24 | 67.28     | 42.69  | 88.11    |
> |                  |       |           |        |          |
> | gpt2-ft-VR       | 31.70 | 25.63     | 41.52  | 72.75    |
> | gpt2-large-ft-VR | 31.99 | 34.11     | 30.12  | 80.50    |
> | gpt2-xl-ft-VR    | 43.95 | 48.25     | 40.35  | 84.33    |
> | llama-ft-VR      | 53.92 | 77.90     | 41.23  | 89.27    |
> | llama2-ft-VR     | 49.47 | 91.34     | 33.92  | 89.45    |
>
> We can see a largest drop of performance across all finetuned models, which is consistent with our observation in the paper. And also variable refactorization to swap the variable names causes a larger drop than paraphrasing, which also aligns with our observation in the paper.
>
>
>
> Thank you again for all your comments, and don't hesitate to let us know if you have any further questions.

---

### Official Review · Reviewer_iTjK · 2023-11-05

**Soundness:** 2 fair
**Presentation:** 4 excellent
**Contribution:** 3 good
**Rating:** 6
**Confidence:** 5

**Summary:**

This paper presents a large-scale benchmark of causal reasoning capabilities of large language models.  This benchmark is constructed in an attempt to avoid many of the threats to the validity of previous causal LLM benchmarking approaches.  For example, it avoids reliance on numerical-data processing to focus on verbal statements; and avoids text that may have been included in the LLM's training data.  The paper uses the created benchmark set to evaluate the causal reasoning performance of a large number of open source, and proprietary LLMs, including GPT-4, and also runs experiments with fine-tuned versions of selected models.  The benchmark results indicate that LLMs do not perform causal reasoning.

**Strengths:**

- The construction of the Corr2Cause benchmark is a strong attempt to disentangle "pure causal inference" capabilities of language models from potential construct validity threats.  For example, avoiding potentially memorized scenarios, and avoiding a requirement of numerical analysis of existing data-driven benchmarks.

- The Corr2Cause benchmark is evaluated on a broad set of language models.  The additional experiments on fine-tuned models provide additional insight.

**Weaknesses:**

- LLM performance in many tasks is strongly sensitive to system prompt instructions framing the LLM's role and general task.  Appendix A seems to indicate that there was no such "system prompt" used.  The paper and its findings would be strengthened if it explored whether such instructions (such as stating the rules of causal inference or including few-shot examples) would help performance.

- Similarly, different kinds of structured reasoning (e.g., chain of thought) have been found to improve performance significantly --- and also to sometimes give insight into why failures might be occurring, e.g., by verbalizing mistaken assumptions.  The prompt in Appendix A seems to only ask for a direct Yes/No answer, however.

- What do we learn from studying "pure causal inference" vs other facets of causal reasoning? The paper clearly has a strong perspective on what it would mean for a LLM to perform causal reasoning, naming its chosen task "pure causal inference".  The paper would be greatly strengthened by discussing this perspective in more depth.

**Questions:**

I appreciate the task the authors are addressing, and the approach they have chosen.  The major weakness of the paper I am concerned about are those relating to how the models are prompted to improve LLM behavior (lack of system prompt, few shot examples, chain of thought reasoning) and error analysis of results (e.g., perturbing inputs, running shap analysis).  That is what gives me the most pause in interpreting the results of these benchmarks---would the LLM have succeeded with different prompting strategies?

Below, I also list a few additional questions, more about the broader framing of the corr2cause benchmark.

- Some work has found differential abilities/deficiencies in LLM's symbolic reasoning vs semantic reasoning [https://arxiv.org/abs/2305.14825].  Corr2Cause seems to rely heavily on an assumption that the LLM is able to reason symbolically.  Noting the "natural stories" variant proposed in sec 4.6 may address this, do the authors see this as a potential concern for interpreting the results of the benchmark results in the paper?

- Why is "pure causal inference" the right task for evaluating the causal capabilities of large language models? For example, the formal reasoning approaches describes in Section 2 that guide the creation of Corr2Cause are relatively new inventions in human history.  But I think we'd hesitate to say that people could not reason causally before, e.g., DAGs were invented.  This seems like a good opportunity to clarify what underlying functionality we are trying to achieve and how pure causal inference vs other causal reasoning tasks relate to broader goals and purposes.  And to be perfectly clear, I think this is a fine task to evaluate but would like to learn more about the authors' perspectives on relative importance, etc. of testing different facets of causal reasoning.

- A deeper discussion of corr2cause in the context of reasoning and inference tasks would be interesting, especially in regards to design choices made in this benchmark as compared to others.  How were those other benchmarks designed and how does that influence the design of corr2cause?

---

> ### Author Response · Authors · 2023-11-23
>
> Thank you for the overall positive feedback on our paper, particularly regarding the novel and intriguing nature of the Corr2Cause task, and its evaluation across a diverse set of language models.
>
> ### Optimizing LLM Performance
>
> > I appreciate the task the authors are addressing, and the approach they have chosen. The major weakness of the paper I am concerned about are those relating to how the models are prompted to improve LLM behavior (lack of system prompt, few shot examples, chain of thought reasoning) [...] would the LLM have succeeded with different prompting strategies?
>
> We are grateful for your insightful query, especially regarding the optimization of prompting strategies when reporting LLM performance. Reflecting on this, we see merit in both our paper's approach, which evaluates LLMs based on plain queries that a typical user might pose, and your suggestion, which explores the potential for maximizing performance with tailored prompts.
>
> In response, we conducted new experiments incorporating the three strategies you mentioned: system prompts, few-shot examples, and chain-of-thought reasoning. We present the results as follows (and will keep adding new experiments to extend to more models and different combinations of the strategy):
>
> | Model                           | F1    | P | R | Acc |
> | ------------------------------- | ----- | --------- | ------ | -------- |
> | GPT-3.5 (Plain Query; Original) | 21.69 | 17.79     | 27.78  | 69.46    |
> | GPT-3.5 (Enhanced Query)        | 25.44 | 17.29     | 48.11  | 52.01    |
>
> The enhanced query is achieved by combining all three strategies, which is 4 points better on F1 than the original performance. However, we can see that despite adding all three strategies, GPT-3.5 still does not do well at this challenging Corr2Cause task. As for experimental details, for the enhanced query, we (1) use a system prompt to indicate the expertise ("You are a highly intelligent question-answering bot with profound knowledge of causal inference."); (2) include two few-shot examples, one positive, one negative; and (3) add the chain-of-thought prompting with "Let's think step by step.") to induce language models to produce step-by-step reasoning. Also, since the initial experiments were conducted earlier this year, we rolled back to the closest previous GPT checkpoints in March.
>
> ### Significance of Pure Causal Reasoning
>
> > Why is "pure causal inference" the right task for evaluating the causal capabilities of large language models? For example, the formal reasoning approaches describes in Section 2 that guide the creation of Corr2Cause are relatively new inventions in human history. But I think we'd hesitate to say that people could not reason causally before, e.g., DAGs were invented. This seems like a good opportunity to clarify what underlying functionality we are trying to achieve and how pure causal inference vs other causal reasoning tasks relate to broader goals and purposes. And to be perfectly clear, I think this is a fine task to evaluate but would like to learn more about the authors' perspectives on relative importance, etc. of testing different facets of causal reasoning.
>
> Thank you for this thoughtful question and your openness to discussion. Our proposal for the Corr2Cause task is inspired by interdisciplinary insights, particularly from philosophy, where reasoning is categorized into pure (rationalism) and empirical (empiricism) reasoning. The latter, empirical reasoning, is usually framed as commonsense reasoning in existing NLP literature, which corresponds to your intuition that people can reason before, e.g., DAGs were invented.
>
> In contrast, pure reasoning, or *a priori* reasoning, involves logic and abstract thought, independent of sensory experience. In the causal domain, we believe that the necessity of pure causal reasoning is analogous to that of math, which aid human understanding independent of sensory experience. As you mentioned the natural story realizations in our Appendix B, we highlight that our symbolic Corr2Cause reasoning pattern holds regardless of the instantiation of the variables, which makes us rely on these inference rules *independent of our experience*, or even *overcome cognitive biases* that confuse correlation with causation.
>
> Upon reviewing existing NLP datasets on causal reasoning, we found a predominant focus on empirical commonsense reasoning, with little to no emphasis on pure reasoning in the literature. Recognizing the longstanding philosophical value of pure reasoning, which dates back to early Greek philosophers like Parmenides and Pythagoras and is further developed by figures like Plato, Descartes, and Kant, we believe in its significant potential to enrich human knowledge beyond empirical experiences.
>
> Both rational and empirical forms of knowledge are crucial, complementing each other in expanding the realm of human understanding. This goal is what we aim to highlight and explore through the Corr2Cause task.

---

> > ### Author Response · Authors · 2023-11-23
> >
> > > A deeper discussion of corr2cause in the context of reasoning and inference tasks would be interesting, especially in regards to design choices made in this benchmark as compared to others. How were those other benchmarks designed and how does that influence the design of corr2cause?
> >
> > We appreciate your suggestion for a deeper discussion on the design choices of Corr2Cause in comparison with other benchmarks. We have conducted a comprehensive survey of existing causal reasoning datasets and found that most datasets fall into two categories: commonsense causality, which heavily relies on empirical experiences; and an information extraction framing to identify cause and effect from text, which is more of a linguistics task. We provide a summary table of the existing body of literature below.
> > |                                                              | Pure Causal Reasoning | Commonsense Causality/Empirical Causal Reasoning | Causal Relation Extraction (Relying on Linguistics) |
> > | ------------------------------------------------------------ | --------------------- | ------------------------------------------- | --------------------------------------------------- |
> > | **Type 1**                                                   |                       |                                             |                                                     |
> > | COPA [(2012)](https://people.ict.usc.edu/~gordon/copa.html)  | ✗                     | ✓                                           | ✗                                                   |
> > | Event2Mind [(2018)](https://uwnlp.github.io/event2mind/)     | ✗                     | ✓                                           | ✗                                                   |
> > | ATOMIC [(2019)](https://allenai.org/data/atomic)             | ✗                     | ✓                                           | ✗                                                   |
> > | SocialIQA [(2019)](https://allenai.org/data/socialiqa)       | ✗                     | ✓                                           | ✗                                                   |
> > | TimeTravel [(2019)](https://paperswithcode.com/dataset/timetravel) | ✗                     | ✓                                           | ✗                                                   |
> > | Goal-Step [(2020)](https://aclanthology.org/2020.emnlp-main.374/) | ✗                     | ✓                                           | ✗                                                   |
> > | Abductive (ART) [(2020)](https://paperswithcode.com/dataset/art-dataset) | ✗                     | ✓                                           | ✗                                                   |
> > | Com2Sense [(2021)](https://aclanthology.org/2021.findings-acl.78/) | ✗                     | ✓                                           | ✗                                                   |
> > | CRASS [(2022)](https://aclanthology.org/2022.lrec-1.229/)    | ✗                     | ✓                                           | ✗                                                   |
> > | **Type 2**                                                   |                       |                                             |                                                     |
> > | SE2021 Task8 [(2010)](https://paperswithcode.com/dataset/semeval-2010-task-8) | ✗                     | ✗                                           | ✓                                                   |
> > | EventCausality [(2011)](https://aclanthology.org/D11-1027/)  | ✗                     | ✗                                           | ✓                                                   |
> > | Causal-TimeBank [(2014)](https://github.com/paramitamirza/Causal-TimeBank) | ✗                     | ✗                                           | ✓                                                   |
> > | CaTeRS ([2016](https://paperswithcode.com/paper/caters-causal-and-temporal-relation-scheme)) | ✗                     | ✗                                           | ✓                                                   |
> > | BECauSE ([2017](https://aclanthology.org/W17-0812/))         | ✗                     | ✗                                           | ✓                                                   |
> > | TellMeWhy [(2021)](https://aclanthology.org/2021.findings-acl.53/) | ✗                     | ✗                                           | ✓                                                   |
> > | **Type 3**                                                   |                       |                                             |                                                     |
> > | **Corr2Cause Inference (Ours)**                              | ✓                     | ✗                                           | ✗                                                   |
> >
> > Thank you again for all your comments, and please don't hesitate to let us know if you have any further questions.

---

### Meta-Review · Area_Chair_zUhP · 2023-12-09

**Metareview:**

The discussion and feedback provided by the reviewers have led to a consensus that the paper presents a novel dataset and task, which is an interesting attempt at LLM's causality. The reviewers appreciate the attempt to disentangle pure causal inference capabilities of language models and the extensive experiments conducted. However, there are questions about the prompting strategies used, the depth of evaluation of the causal reasoning, and the benchmark's construction and difficulty level.

The authors showed their efforts to address the reviewers' concerns by conducting additional experiments and providing further clarification. Some reviewers have increased the score to 6.

AC recognizes the value of the work and suggests that the paper could potentially inspire future research in causal inference for large language models. It is recommended that the authors refine the work by considering the feedback in depth and submit an improved version in camera-ready.

**Justification For Why Not Higher Score:**

Although the paper shows an interesting potential, it lacks more insights and rigorous experimental designs.

**Justification For Why Not Lower Score:**

A clear and interesting paper overall. Should be attractive to a broad ICLR audience.

---

### Decision · Program_Chairs · 2024-01-16

Accept (poster)